# The Role of Active Packaging in the Defense Against Foodborne Pathogens with Particular Attention to Bacteriophages

**DOI:** 10.3390/microorganisms13020401

**Published:** 2025-02-12

**Authors:** Rajesh V. Wagh, Ruchir Priyadarshi, Ajahar Khan, Zohreh Riahi, Jeyakumar Saranya Packialakshmi, Pavan Kumar, Sandeep N. Rindhe, Jong-Whan Rhim

**Affiliations:** 1Department of Livestock Products Technology, College of Veterinary Science, Guru Angad Dev Veterinary and Animal Sciences University, Ludhiana 141004, Punjab, India; vetpavan@gmail.com; 2BioNanocomposite Research Center, Department of Food and Nutrition, Kyung Hee University, Seoul 02447, Republic of Korea; ruchirpriyadarshi@gmail.com (R.P.); arkhan.029@gmail.com (A.K.); zohreh.riahi95@gmail.com (Z.R.); saralakshmi8@gmail.com (J.S.P.); 3Department of Livestock Products Technology, College of Veterinary and Animal Sciences, Maharashtra Animal and Fishery Sciences University, Parbhani 431402, Maharashtra, India; sandeeprindhe@gmail.com

**Keywords:** active food packaging, phage therapy, antimicrobials, food safety

## Abstract

The increasing demand for food safety and the need to combat emerging foodborne pathogens have driven the development of innovative packaging solutions. Active packaging, particularly those incorporating antimicrobial agents, has emerged as a promising approach to enhance food preservation and safety. Among these agents, bacteriophages (phages) have gained significant attention due to their specificity, efficacy, and natural origin. This manuscript explores the role of active packaging in protecting against foodborne pathogens, with a particular focus on bacteriophages. The review overviews recent advances in antimicrobials in food packaging, followed by a detailed discussion of bacteriophages, including their classification, mode of action, multidisciplinary applications, and their use as antimicrobial agents in active food packaging. The manuscript also highlights commercially available bacteriophage-based products and addresses the challenges and limitations associated with their integration into packaging materials. Despite their potential, issues such as stability, regulatory hurdles, and consumer acceptance remain critical considerations. In conclusion, bacteriophages represent a promising tool in active packaging for enhancing food safety, but further research and innovation are needed to overcome existing barriers and fully realize their potential in the food industry.

## 1. Introduction

Foodborne illnesses are usually infectious or toxic in nature and caused by bacteria, viruses, parasites, or chemical substances entering the body through contaminated food [1]. These illnesses are usually called foodborne infections or food poisoning [2,3]. Lifestyle shifts and altered food consumption patterns contribute to rising food poisoning incidents. It is documented that 31 pathogens cause 37.2 million illnesses yearly in the US. Of those, 9.4 million are foodborne: 59% are caused by viruses, 39% by bacteria, and 2% by parasites [4,5]. Norovirus is one of the infections causing the most illness (58%), followed by non-typhoidal *Salmonella* spp. (11%), *Clostridium perfringens* (10%), *Campylobacter* spp. (9%), *Staphylococcus aureus*, *Clostridium botulinum*, *Listeria* spp., *Escherichia coli*, and *Vibrio* [1,2,4]. Bacterial foodborne illnesses pose significant challenges due to their diverse nature, evolving antimicrobial resistance, and complex transmission pathways.

Antimicrobial resistance (AMR) is a critical global health threat [6] and is no longer a future prediction. It can impact anyone, causing severe health issues. Despite advancements in food safety, foodborne illnesses are increasing. This urgency has driven collaboration between industry and science to find solutions [7,8,9,10]. The CDC (Centers for Disease Control and Prevention) reports that four simple steps, i.e., Clean, Separate, Cook, and Chill, can substantially control foodborne illness and help control household-level food-related disease [10]. But as new generations of microbial threats, including AMR, evolve, consumer demands are driving scientists and the food industry toward strategies that improve the shelf life, real-time monitoring, sensory, packaging, and overall quality attributes of food [11,12,13].

Collaborative efforts have yielded strategies to prevent microbial contamination throughout the food supply chain [14]. Biopolymer packaging, particularly active packaging incorporating antimicrobial agents, offers promising solutions [13]. Active packaging, utilizing biopolymers with embedded antimicrobials like enzymes, bacteriocins, essential oils, and nanoparticles, provides cost-effective, environmentally friendly, and effective food safety enhancements [15,16].

Recently, bacteriophages have emerged as a compelling functional strategy. These ecological and sustainable nanotools selectively target foodborne pathogens while leaving beneficial microbes unharmed [13]. As early as 1896, Ernest Hanbury Hankin demonstrated the presence of bacteriophages in purified water from the Ganges and Jamuna rivers in India. His report, published in the *Archives of the Pasteur Institute*, showed that these bacteriophages exhibited bactericidal activity against *Vibrio cholerae* [17,18]. In 1915, Frederick Twort documented the antibacterial effect of bacteriophages while studying the growth patterns of vaccine viruses in cell-free culture media [19]. Two years later, the French microbiologist D’Herelle of the Pasteur Institute studied the intestinal bacteria of dysentery patients and discovered an irreversible antagonist that retained its bactericidal activity even after passing through a filter paper, which he identified as a virus and named phage. As implied, the virus destroyed the bacterial cell. Therefore, D’Herelle often called it an “immune microbe” [19]. Nevertheless, with the advent of antibiotics, phages were largely overlooked in mainstream research. Research conducted in the 1980s revealed that bacteriophages outperformed antibiotics in inactivating *E. coli* in mice [18,19,20].

Innovative bacteriophages are increasingly utilized as antimicrobials across diverse sectors, including healthcare, food microbiology, and packaging [16,17,21,22,23]. However, there seems to be a lack of comprehensive literature review on bacteriophages and their application as active ingredients in biopolymer-based food packaging materials.

This review aims to provide a detailed, comprehensive overview of bacteriophages and their classifications while also exploring recent advancements in active antimicrobial agents for bio-based food packaging. A significant focus is placed on the emerging field of bacteriophage-reinforced packaging films, which address the critical global challenge of antimicrobial resistance.

## 2. Antimicrobials in Food Packaging

Active packaging assisted by antimicrobial agents is a promising approach to protect packaged foods from deterioration during the supply chain that may occur due to contamination by microorganisms (bacteria, viruses, parasites), which can cause foodborne illness [21]. Food packaging with antimicrobial additives is a type of active packaging that utilizes antimicrobial agents, antimicrobial carriers, and growth inhibitors [22]. Recently, researchers around the world have been interested in developing technologies to minimize the growth of foodborne pathogens through active food packaging [23,24]. This type of packaging can be developed by encapsulating the antimicrobial agent in the packaging material. The antimicrobial agent’s effectiveness is mainly determined by its ability to migrate to the food’s surface, the primary site of spoilage. Encapsulating the additive in the film allows the antimicrobial agent to gradually diffuse to the food surface, thereby extending its activity. However, this diffused concentration is very small but high enough to cause an antimicrobial effect. This approach eliminates the need to add these compounds to the food that consumers prefer directly [25]. Therefore, antimicrobial food packaging helps reduce food spoilage by inhibiting or delaying microbial growth, maintaining the quality, appearance, flavor, and nutritional value of the packaged product, and preserving food for a longer period [26].

Due to their high surface-to-volume ratio, enhanced surface reactivity, and physicochemical and antimicrobial properties, nanomaterials are more effective in inhibiting pathogen activity than their microscopic or macroscopic counterparts. Incorporating or nano-encapsulating natural antimicrobial nanocarriers has improved the performance and efficacy of active food packaging [27]. There are a number of antimicrobial materials that have been discussed for use in food safety or food packaging applications so far, which can be categorized into (i) inorganic materials such as metals and metal oxides, (ii) organic materials such as polymers, enzymes, and organic acids, (iii) biomass-derived essential oils such as thyme, lemon verbena, oregano, pimento, citron, clove, cypress leaf essential oils, and lemon balm, (iv) peptides such as lactoferrin and nisin, (v) plant extracts such as bearberry, rosemary, grape seed, green tea, sage plant extracts, ginger, and garlic, and (vi) pure biologically active ingredients such as carvacrol, thymol, and caffeic acid phenethyl ester [28,29]. Among antimicrobial agents, metal nanoparticles (MNPs), carbon nanomaterials, metal oxides, and essential oils loaded in biopolymer matrices are widely used in active packaging to meet antimicrobial efficacy. In addition to antimicrobial additives, biopolymer chitosan also has excellent antimicrobial activity. These multifunctional films inhibit the growth rate of foodborne pathogens, extend the shelf life (storage) of packaged foods, and are considered potential antimicrobial biopolymer materials for food packaging [24].

The antimicrobial activity of biopolymer-based films is determined by several parameters such as molecular weight, concentration, blended additives, reaction conditions, and types of microorganisms. For example, Priyadarshi et al. [30] studied chitosan-based functional films loaded with *Prunus armeniaca*-based essential oil, AKEO, for the active packaging of bread. They reported that chitosan films loaded with AKEO exhibited excellent antimicrobial activity against *B. subtilis* and *E. coli* bacterial strains. No mold growth was found in bread packaged with chitosan films loaded with AKEO [30]. Also, chitosan/ZnO coating significantly inhibited the growth of *E. coli* on white-brined cheese from 4 °C to 10 °C [31]. In addition, encapsulating clove oil (antimicrobial agent) in ZnO-reinforced gelatin films can inhibit microbial spoilage, maintain quality, and improve the shelf life of packaged shrimp [32]. Adding silver and zinc oxide to chitosan can effectively increase the antibacterial effect of composite films [33,34]. Pullulan/chitosan-based functional films reinforced with propolis and ZnO showed a 100% bactericidal effect against *L. monocytogenes* and *E. coli* after 12 h of exposure [34]. Pullulan/chitosan films containing propolis and ZnO were observed to have potent antimicrobial and antioxidant effects (active roles) on pork packaging, thereby helping to inhibit lipid peroxidation and extend the shelf life of the packaged meat [35]. CMC films coated with silver nanoparticles (AgNPs) and cellulose acetate films loaded with AgNPs exhibited potent antibacterial activities against *E. coli* and *Staph. aureus* [36,37]. The incorporation of caffeine phenethyl ester nanoparticles into cellulose-based films effectively reduced microbial growth in packaging applications [38]. Ezati et al. reported that nitrogen-doped glucose carbon dots (NGCDs)-integrated cellulose nanofiber (CNF)-based films exhibited potent antibacterial activities against bacteria (*L. monocytogenes* and *E. coli*) and mold (*A. flavus*) [39]. The CNF/NGCD films showed distinctive bactericidal effects against both bacterial strains depending on the exposure time (Figure 1a). The CNF/NGCD film also exhibited excellent antifungal activity by completely inhibiting the growth of *A. flavus* after two days of culture (Figure 1a). SEM micrographs of CNF-based film-treated microorganisms verified the bactericidal effect of the manufactured films. As shown in Figure 1b, carbon dot interactions altered the morphology of the microorganisms and caused severe cell membrane destruction [39].

Additionally, the composition of poly(vinyl alcohol) (PVA)/cinnamon essential oil (CEO)/β-cyclodextrin (β-CD) nanofiber films exhibited the best antibacterial abilities against Gram-positive and Gram-negative bacteria [36]. The composite films efficiently extended the storage efficiency of strawberries and showed potential for antibacterial packaging applications [40]. Rieger and Schiffman studied the antibacterial ability of electrospun chitosan/cinnamaldehyde/poly(ethylene oxide) nanofibers against *E. coli* [41]. They reported the unique antibacterial efficacy of chitosan associated with the rapid release of cinnamaldehyde, which resulted in increased inactivation rates against *P. aeruginosa* and *E. coli* [41]. Table 1 summarizes the recent studies on biopolymer-based antibacterial packaging materials/activators for food applications.

## 3. Bacteriophages

Obligate, ubiquitous intracellular parasites (microorganisms of the virus family) that often infect and kill bacteria are called bacteriophages or phages [13,19]. The term “bacteriophage” comes from bacteria, and “phagein” is a Greek word meaning to eat/devour. [17]. Phages are widespread and are abundant in water, plants, soil, and animals that have bacterial hosts. In addition to their target bacterial hosts, bacteriophages are harmless to most other organisms, including humans, thereby maintaining the balance of the microbial host. Phages take over the host’s biosynthetic mechanisms and begin to replicate [20,57,58].

Over 10^32^ bacteriophage types have been identified, and this number continues to grow as research progresses. Over 6000 have been described morphologically, including variations in their capsid (head) structure, tail structures, and the presence or absence of envelopes [47,48,49,50,58].

Figure 2 illustrates the classification of bacteriophages based on their morphological and genomic characteristics and highlights the major families. It also provides information on the typical hosts for each family, such as bacteria from various genera, and includes specific examples of well-known bacteriophages within each family.

### 3.1. History of Bacteriophages

As discussed earlier, the frequent occurrence of bacteriophages within the environment underscores their significant ecological role and contributes to their abundance. A variety of phages are observed in most biological entities in an ecosystem [59]. Like bacterial populations, phages are ubiquitous wherever bacteria are found and sometimes overwhelm bacterial populations. Phages can be easily isolated from soil, human and animal waste, sewage, plants, and animals. The largest viral populations are found in marine habitats such as marine sediments, near-sea surface sediments, and seawater [60]. Polymerase chain reaction (PCR) analysis confirmed the presence of bacteria and archaeal phages in 14th-century fecal samples. The estimated genome size of bacteriophages ranges from 2.4 to 735 kb. The size of phages ranges from 20 to 300 nm [55,56,57,61]. Morphologically, phages exhibit a strong geometric symmetry. Most phages have a significant tail or tail-like structure. They are generally composed of three main structures: a head, a capsid, and a tail. Capsomeres (protein subunits) make up the head or capsid. A collar connects the head to the capsid. The nucleic acids are relatively short and can encode only a few genes like structural genes (capsid, tail and collar proteins) and lysogenic and lytic genes [60]. The number of genes is directly proportional to the host cell’s dependency on the phage [62,63].

### 3.2. Classification of Phages

Since phages were identified in the early 18th century, various attempts have been made to classify them. In the late 18th century, the French botanist Adason attempted to classify phages by considering nearly 60 unweighted parameters. This classification was short-lived and refined and modernized with the advancement of science and technology. However, there is no unified classification system to date [64]. Viruses that infect eubacteria and archaea are generally classified as “prokaryotic viruses” [64,65]. Early classifications used serology, particle size, host range, morphology, pathogenicity, and stability. Then, in 1966, the International Committee on Taxonomy of Viruses (ICTV) established a classification system based on the characteristics of the virus and its nucleic acid, which is still used today [66]. ICTV utilizes about 70 characteristics of phages for classification, with special reference to the characteristics of nucleic acids. Nucleic acids are used as a subsequent parameter in morphology-based classification to overcome uncertainty. Depending on the shape, phages are tail-shaped, polyhedral, filamentous, and polymorphic [50,67]. Depending on the genome classification factor, they can be linear, circular, or single- or double-stranded DNA or RNA.

Tailed phages range from 30 to 110 nm and are the most commonly studied. They are step synonymous and the oldest viruses. These virus particles have an icosahedral or elongated head without an envelope [67]. The nucleic acid is a linear double-stranded DNA molecule with a genome size of 8–500 kbp. Tailed phages have a high guanine and cytosine content (up to 72%) and are highly homologous to their host bacteria. Tailed phages belong to the order Caudovirales. This sequence contains most viruses, accounting for 10^31^ tailed phages in the ecosystem or 6% of all viruses [65]. This order is further subdivided into the families *Myoviridae* (phages with complex contractile tails), *Podoviridae* (phages with non-contractile short tails), and *Siphoviridae* (phages with non-contractile long tails), which are the most abundant tailed phages.

The virus particle contains 7–49 structural proteins and a portal protein. The spiral tail is a protein tube with structures such as a basal plate, spikes, and fibers for host cell interaction [68]. Despite the tail, most phages are not mobile and move from one place to another through physical movements such as Brownian motion [69]. The remaining 4% of viruses fall into three morphological categories: polyhedral, filamentous, or polymorphic phages. Cuboid or polyhedral viruses include both DNA and RNA viruses.

Polyhedral or cuboid viruses include seven prokaryotic viruses, five well-classified families, and two unclassified families [64,65]. The five classified families are *Microviridae*, *Corticoviridae*, *Tectiviridae*, *Leviviridae*, and *Cystoviridae* (Figure 2).

The family *Microviridae* includes non-enveloped, spherical DNA with single-stranded viruses with icosahedral symmetry [70]. The *Microviridae* family is divided into the genus Microvirus, which infects enteric bacteria, and three genera in the subfamily *Gokushovirinae*, which feed on parasitic *Bdellovibrio*, *Chlamydia*, and *Spiroplasma*. They also differ in genome size. Microvirus genomes range from 5.3 to 6.1 kb, while *Gokshovirus* genomes are smaller (4.4 to 4.9 kb) [70,71].

Phages in the family *Corticoviridae* are viruses with icosahedral symmetry that contain dsDNA and an internal membrane [72]. The family encompasses one genus containing one species. The virions host the Gram-negative bacteria *Pseudoalteromonas*. The nucleic acid is a highly coiled DNA with numerous negative supercoils. The molecular size of the coiled DNA is 10 kb, encoding for 21 proteins enclosed inside the internal membrane and an outer protein capsid facilitating their multiplication via rolling circle replication commenced by protein P12 [72].

The *Tectiviridae* family includes highly virulent Enterobacterial phages. They are tailless, double-stranded DNA viruses with no envelope and icosahedral symmetry [64]. Approximately 15 kb of linear DNA is coiled inside an inner membrane formed by lipids and viral-encoded proteins. These phages host actinobacteria such as *Enterobacteria*, *Bacillus* sp., thermophilic bacteria *Thermus* spp., and *Streptomyces* sp. [73]. WheeHeim and Forthebois are newly added members, i.e., the *Tectiviridae* family [74]. These viruses are preferred biological control agents for controlling *Streptomyces scabiei*, a common scab pathogen that causes significant economic losses worldwide [75].

*Fiersviridae*, formerly *Leviviridae*, are Polio virus-like phages (Figure 2). The phages are spherical viruses with linear single-stranded RNA. These are non-enveloped viruses with icosahedral symmetry affecting *E. coli* and *Pseudomonas*. The *Fiersviridae* consists of two genera, *Allolevivirus* and *Levivirus*. These RNA phages serve as models for RNA assembly. The genome has four different proteins that encode for the coat, replicase enzyme, maturation, and protein lysis [76].

The Cystovirus is the only genus in the family of *Cystoviridae*. Taxonomically, the genus *Cystovirus* has seven species [64]. These are enveloped virions with tripartite dsRNA circumscribed inside icosahedral spiked protein shells. The inmost protein polymerase layer promotes genome packaging, replication, and transcription. These phages are pathogens of Gram-negative plant bacteria, *Pseudomonas syringae*, and some human pathogens. The linear dsRNA has three segments, with each segment size ranging from 2.9 to 6.4 kb. These segments encode for a polycistronic mRNA translated by prokaryotic translation machinery [77]. Serpentine Lake Hispanica (SHI) and the Sulfolobus turreted icosahedral virus (STIV) are two unnamed members of the cubic bacteria family. SHI is the only isolated and unidentified member that contaminates/infects halobacteria. The genome is a linear dsDNA. These virions have lipids and structures similar to *Tectivirus* [78,79]. The STIV is the sole representative of the STIV. These viruses host *Sulfolobus*, a hyperthermophilic archaeon [59,80].

The next category of prokaryote viruses is filamentous phages. The filamentous phages are DNA viruses that include three families: *Rudiviridae*, *Inoviridae*, and *Lipothrixviridae*.

The phages of *Inoviridae* are structurally longer than the host bacteria, varying from 800 nm to 4 µm. In-depth consideration of filamentous phages comes from comparative studies of the archetypal *E. coli* phages, the Ff phages. Though Ff phages are not representatives of the family of filamentous phages, their characteristic features are collectively similar [69]. The nucleic acid of *Inoviridae* is single-stranded, but due to the rolling circle model of replication, dsDNA is produced as an intermediate. These are lysogenic phages; hence, host cells remain structurally intact. These phages affect *Enterobacter*, *pseudomonads*, *xanthomonads*, *vibrios*, and *mycoplasmas* (Figure 2).

The remaining two families of this morphological group are *Lipothrixiviridae* and *Rudiviridae. Lipothrixiviridae are* rod-shaped enveloped phages with linear dsDNA and two DNA-binding proteins. The envelope comprises viral proteins and lipids (host-derived). The host range is limited to the hyperthermophilic archaebacteria *Thermoproteus tenax*. These are mostly lytic phages [81,82]. *Rudiviridae* is a double-stranded DNA virus with a genome size ranging from 24,655 to 35,482 bp. Morphologically, *Rudiviridae* virions are non-enveloped straight rods. These viruses are thermophilic bacteria that inhabit thermal springs where temperatures exceed 80 °C and the pH is below 3 [60,83].

The last morphologically distinct group is the pleomorphic phages encompassing various shapes. There are seven families: *Fuselloviridae*, *Plasmaviridae*, *Guttaviridae*, *Globuloviridae, Ampullaviridae*, and *Bicaudaviridae* (Figure 2). All the members of pleomorphic phages have double-stranded DNA as the nucleic acid [84].

The *Plasmaviridae* family of this group is pathogens of mycoplasmas only and has one member, *Acholeplasma* phage L2. As mycoplasmas are devoid of a cell wall, the viruses are infected by membrane fusion. These are non-capsid viruses with an envelope and nucleoprotein [81,84,85].

The family *Fuselloviridae* has lemon-shaped surface morphology with heterogeneous sizes (50 to 100 nm). The thermophilic archeon *Sulfolobus* spindle-shaped virus-1, a type species of genus *Fusellovirus* belonging to this family, are larger phages with particle lengths of about 300 nm. Their nucleic acid is a positively supercoiled double-stranded DNA with polyamines and basic proteins [82]. *Salterproviruses* are spindle-shaped with a long tail. Two known members of these genera are His1 and His2, both isolated from *Haloarcula hispania*, an archeon isolated from hypersaline lagoons [78]. These virions are stable only if maintained at high salt concentrations. The genome is a linear dsDNA. They are lytic phages but can exist in an unstable carrier state. Members of this genus can be restricted by terminal proteins and DNA polymerase [78].

The members of the family *Guttaviridae* host *Sulfoloubus* [64]. The virions of the family are ovoid-shaped. These are viruses (enveloped) with a novel and unique beehive-like surface pattern with projections masked by a “beard” of long fibers. Engaging DNA facilitates transcription of the heavily methylated genome of this family as a template [82]. The virions of *Ampullaviridae* infect the archaebacteria of the genus *Acidianus*. These virions have the shape of a bottle with smooth ends. The envelope encloses the funnel-shaped cores of the virions. This family of virions, genome replication, is facilitated by the family B-DNA polymerase primed by virus-encoded proteins. Ampullavirus has three species: *Bottigliavirus* ABV, *Bottigliavirus* ABV2, and *Bottigliavirus* ABV3 [64]. *Bicaudaviridae* is the largest of all the archeal phages, infecting *Acidianus* like the phages of the *Ampullaviridae* family. These are spindle-shaped phages with helical nucleocapsids. The virions have two tails, one at each pointed end. The virions have the largest genome circular double-stranded DNA of size 62,730 bp [81].

The last family of phages, *Globuloviridae*, is an archeal phage that infects the phylum *Thermoproteota* (genera: *Pyrobaculum* and *Thermoproteus*). These are spherical and lipid-containing enveloped viruses of size 100 nm [86].

The ICTV added several new families by reclassifying the viruses based on nucleic acid. Currently, in ssDNA phages, *Plectroviridae* and *Finnlakeviridae* are added to the list. In dsDNA phages, *Autographiviridae*, *Chaseviridae*, *Drexleviridae*, *Herelleviridae*, *Ackermannviridae, Podoviridae*, *Sphaerolipoviridae*, *Demercviridae*, and *Plasmaviridae* are added to the list [60,64]. With advancements in scientific technologies and extensive research, a deeper understanding of viruses is being achieved, and novel phages belonging to various families are still being identified, making the long list of viruses open for inclusion.

### 3.3. Virus Replication

Bacteriophages exhibit two distinct infection cycles: lytic (virulent) and lysogenic (temperate) [87]. In the lytic cycle, following phage adsorption to host receptors, an eclipse phase occurs where infectious virions are undetectable. This latent period encompasses adsorption to the release of new virions, leading to host cell lysis [88]. In the lysogenic cycle, the phage genome integrates into the host bacterial chromosome, replicating passively with the host [20]. However, certain signals can induce the integrated prophage (lysogen) to enter the lytic cycle. The one-step growth curve describes this characteristic phage replication [64]. While complete separation is impossible, high phage multiplicity generally favors the lytic cycle, reducing the likelihood of lysogeny [89].

## 4. Application of Multifunctional Bacteriophages

### 4.1. Healthcare

Bacteriophages, viruses that specifically target bacteria, offer a promising alternative to traditional antibiotics in combating the escalating global threat of antimicrobial resistance. Phage therapy has demonstrated efficacy in preclinical and clinical trials against a range of bacterial pathogens, including those implicated in wound infections [90,91], respiratory tract infections [92,93], and urinary tract infections [94,95]. Furthermore, the ability to isolate and characterize phages specific to individual bacterial strains provides a personalized approach to infection management, potentially overcoming limitations associated with broad-spectrum antibiotics. Phage therapy continues to thrive in Eastern European countries, including Russia, Georgia, and Poland [59]. The Eliaba Institute of Bacteriophage reported that 95 percent of patients receiving phage therapy showed significant recovery with the least side effects [96].

Figure 3 illustrates various applications of bacteriophages, leveraging their unique properties like host specificity and natural abundance. Phage therapy: This is a prominent application, utilizing phages to combat bacterial infections [97]. Phage therapy offers a potential solution to the growing issue of antibiotic resistance by specifically targeting harmful bacteria. Biocontrol of foodborne pathogens: Bacteriophages can be employed to control foodborne pathogens, ensuring food safety [88,98,99]. Their host specificity and non-toxicity to eukaryotic cells make them a promising alternative to chemical preservatives. Disinfectant products: Phages can be incorporated into disinfectant products, enhancing their effectiveness against bacterial contamination on surfaces and in various environments [100,101]. Agriculture/livestock sector: Bacteriophages can control bacterial infections in livestock and plants, improving animal health and agricultural productivity. Rapid detection methods: Phages can be utilized as indicators for the presence of specific bacteria, enabling rapid detection methods in various settings, such as food safety monitoring and clinical diagnostics [102,103].

### 4.2. Rapid Bacterial Detection

Virus replication and completion of the virus lifecycle are only possible by identifying a suitable and functional host [66,104]. Due to the critical sample preparation and culture requirements, bacteriological methods for pathogen detection are impractical for field use. Transgenic bacteriophages help insert a reporter gene into a bacterial host, where the gene is translated to provide visual cues that help identify the host bacterium (Figure 4).

Typically, a reporter gene expressing a luciferase or fluorescent protein is inserted into the phage genome to provide a fluorescent signal upon translation [58,82]. Several studies have testified to the successful detection of bacterial pathogens using bacteriophages [105,106].

### 4.3. Bacteriophages as Antimicrobial Agents in Active Food Packaging

Bacteriophages are emerging as antimicrobial agents against foodborne pathogens such as *Listeria monocytogenes*, *Bacillus cereus*, *E. coli*, *Salmonella typhimurium*, and *Staphylococcus aureus*. Although limited research has been conducted, there has been a growing trend toward bacteriophage applications in food packaging in recent years. Bacteriophage-based packaging films/coatings were developed for food packaging applications over the past five years and are summarized in Table 2.

According to the literature, one limitation of bacteriophage applications is their lack of stability on the film surface. Various approaches, such as chemical functionalization, encapsulation in polymer composites, bilayer film formation, liposome encapsulation, and phage mixing, have been proposed to increase the stability and efficiency of phage.

Choi et al. [87] developed polycaprolactone (PCL) films containing *E. coli*-specific phage T4. They improved the attachment and distribution efficiency of phage T4 on PCL films by using chemical functionalization instead of physical bonding. They also conducted a meat packaging study to investigate the antimicrobial effect of the fabricated composite films and concluded that covalently immobilized phage exhibited 30 times higher antimicrobial effect than physically adsorbed phage films. As shown in Figure 5, the cross-linking between the succinimidyl ester of PCL and the amine groups of phage increased the adhesion and distribution of T4 phage on the film surface.

Their restricted movement can influence the antibacterial efficacy of immobilized phages. Covalent bonding may limit phage–host interactions, potentially hindering their ability to infect target bacteria. However, careful control of immobilization density through chemical functionalization can optimize antibacterial activity.

In another study, Alves et al. evaluated the effect of ϕIBB-PF7A bacteriophage on the quality attributes of sodium alginate-based films [110]. Cross-linking of sodium alginate films using phage suspension mixed with calcium chloride was performed to increase the phage concentration by 10-fold and prevent phage release. The antibacterial potential of the phase capture films against *P. fluorescens* was studied using contact (bacterial suspension added on top of each film) and immersion methods. The higher phage release rate and available bacteria in the solution of the immersion method improved the antibacterial efficacy of the immersion method by increasing the contact probability between phages and bacteria. Approximately 1.9 and 4.0 log reductions in *P. fluorescens* were observed in the contact and immersion methods, respectively. When the phage-loaded films were applied to chicken fillets, 1-log and 2-log reductions in viable cells were observed after 2 and 5 days, respectively.

Researchers have also found that encapsulating bacteriophages in composite or bilayer films can preserve phage attachment and antibacterial activity for a long time. In this regard, many researchers showed that the entrapment of bacteriophages in biopolymer composite films could effectively stabilize phages. Pure pullulan and whey protein concentrate (WPC) could not preserve A511 phages, but 30WPC/70PULL film protected many phages over 60 days without significantly reducing their antibacterial properties against *L. monocytogenes*. However, some reports have shown that phage microencapsulation of polymer composites is ineffective. Therefore, the behavior of each phase in the selected composition should be investigated. [113].

In another recent study, whey protein concentrate/pullulan composite and poly(lactic acid) (PLA) films with various thickness ratios were used as a phage stabilizer applied to the chicken breast to investigate the antibacterial efficacy. The 30PLA/70WPC was selected as the optimized thickness, retaining the bacteriophage count during sixty storage days, and showed an anti-*Listeria* effect for five days on packed chicken breast fillets [121].

A composite that retains the phase activity for seven months at 24 °C was fabricated by Kimmelshue et al., 2019 [118]. They added *Clavibacter michiganensis* subsp. *nebraskensis* (CN8) bacteriophages to poly(vinyl alcohol) (PVOH), whey protein isolate (WPI), poly(vinylpyrrolidone), poly(methyl vinyl ether), and their blends, and evaluated the phage stability and activity after coating on the maize (*Zea mays* L.) seeds. They reported that polymer properties such as glass transition temperature (T_g_), hydrophilic (functional) groups, and storage temperature had an important influence on bacteriophage attachment efficiency. They pointed out that the T_g_ close to the ambient temperature can increase the phage’s physical stability due to less flexibility. Hydrogen bonding between the polymer (functional) group and maize seed, particularly the outer layer of the cold storage conditions, impart phage stability after storage.

In other related research, different polymer matrices of sodium alginate with gelatin (SAG), sodium caseinate (SC), and polyvinyl alcohol (PVOH) were employed for seizing LISTEX™ P100 bacteriophage to study the matrix effect of the dependence of phage stability on the polymer matrix. They found that phage stability is affected by film formation conditions such as pH, agitation, and drying. For example, the low pH of the sodium alginate film-forming solution can cause an acidic environment, resulting in a reduction in phage stability. The better stability of the phage in the PVOH matrix could be due to the closeness of the phage pH (7.0) to the film-forming solution pH (6.9) [111].

The presence of factors such as acidic compounds, enzymes, and evaporating substances can reduce phage stability and thus deplete its antibacterial activity. Phage encapsulation in liposomes or electrospun nanofibers was introduced as an effective way to increase the functional abilities of phage-based composite films [117], reporting that the encapsulation technique was useful for protecting phases from external influences and improving phage stability and functional properties. In addition, the PVOH coating and nanofiber, including FO1 bacteriophage, were deposited on polyhydroxybutyrate/polyhydroxyvalerate (PHBV) films by solvent casting and electrospinning methods. Only 1-log and 2-log loss in titer value of Felix O1 in the film production process is obtained, demonstrating that the added phage retains its antibacterial properties and the designed films can be used as anti-*Salmonella* packaging films [112].

Alves et al. suggested the separate encapsulation of mixed EC4 and φ135 phages (cocktail) and essential oil (cinnamaldehyde) in the sodium alginate matrix. They observed a synergistic effect that resulted in inhibitory action against *E. coli* and *S. Enteritidis* [110].

Edible coatings containing bacteriophages on foods are a practical tactic to improve bacteriophage efficiency [119]. Radford et al. [116] engineered an edible coating containing two different phages, which provide the stability of the phages and can show antibacterial properties against a wide range of pathogens. [108] prepared gelatin-based composite films with three different phiIPLA-RODI bacteriophage concentrations (5.25 × 10^6^ PFU/mL, 3.48 × 10^6^ PFU/mL, and 1.90 × 10^6^ PFU/mL), and the effect of bacteriophages addition on the physical attributes of films and antimicrobial properties were studied. Their results demonstrated that adding bacteriophages did not change the film’s physical properties, even at the highest concentration of bacteriophages. Cheese slices contaminated with *S. aureus* were used as an in vivo model (food) to test the inhibitory outcome of the fabricated composites (Figure 6). Three methods were chosen to investigate the interactions between bacteriophages and bacteria. These include immersing a piece of cheese in a liquid medium of bacteriophage, coating the cheese sample with a film-forming solution, and wrapping the cheese in a dried film. They found that high-level humidity is required for bacterial infection by the bacteriophages, and the highest antibacterial activity was found in the coated cheese with the film solution (Figure 6), resulting in a stable layer on the cheese retaining the moisture on the sample surface.

Three bacteriophages, vB-EcoM34X, vB-EcoSH2Q, and vB-EcoMH2W, were isolated from the raw beef. The vB-EcoMH2W phage showed a biocontrol effect against several *E. coli* strains. Then, they applied this lytic phage in the chitosan-based edible coating to evaluate their microbial contamination prevention effects in tomato packaging. The data showed that the chitosan matrix could stabilize phage vB-EcoMH2W, and a 3-log reduction in viable *E. coli* cells was detected in the chitosan/bacteriophage-coated samples [109].

In additional studies, the efficacy of adding an O6 phage cocktail to a whey protein concentrate matrix against enteropathogenic and Shiga toxinogenic *E. coli* pathogens present in meat was evaluated. They covered the meat samples with phage-added films and incubated them at 4 ± 1 °C and 24 °C for 24 h, and 37 °C for 1 h. Although the phase showed good stability in WPC during 5 weeks, a suitable phage release pattern was found in liquid and solid systems, which is necessary for antibacterial properties. Non-detectable DH5α and O157: H7 STEC bacterial cell levels were found for the samples at 4 °C and 37 °C, while 10^2^–10^3^ cells survived after 24 h at 24 °C, likely due to the higher percentage of phages released at 4 °C and a lesser self-amplification response at 24 °C as non-optimal bacterial growth conditions [114].

Recently, a new phage JN01 integrated into a gelatin film was developed. The in vitro antimicrobial activity of the films has shown a 1.13 Log CFU/mL decrease in *E. coli* O157: H7 counts after 1 day. When the developed films were used for beef packaging, the *E. coli* count was 1.54 and 1.00 Log CFU/g lower than the gelatin control after 5 and 6 days, respectively. The increased bacteria values after 6 days might be due to the growth of bacteria that have not been in contact with the phages or phage-resistant bacteria. According to the results, the phage JN01 incorporated in the gelatin film displayed an inhibition against *E. coli*, maintained quality, and prolonged the storage life of beef, which could be applied as a packaging material [115].

### 4.4. Commercially Available Bacteriophages

Driven by advancements in biotechnology research, a significant number of government agencies and private companies globally are engaged in the commercialization of bacteriophage-based products. Appendix A provides a comprehensive list of commercially available bacteriophages worldwide, reflecting the progress made in this area.

### 4.5. Challenges and Defects in the Application of Bacteriophage in Packaging Materials

As discussed, bacteriophages have emerged as promising antimicrobial agents with potential applications in food packaging to enhance the safety and shelf life of foods [13]. However, their practical implementation faces several challenges and defects. Firstly, phage stability poses a significant concern [123]. Various environmental factors like temperature fluctuations, humidity, and exposure to UV radiation can inactivate or degrade phage activity [124]. These challenges can be overcome by implanting strategies like encapsulation in protective coatings or polymers (such as chitosan or alginate) and lyophilization (freeze-drying) are employed to preserve their activity. Controlled-release systems have also been developed to ensure that bacteriophages remain effective when needed [125]. Secondly, the narrow host range of most phages limits their effectiveness against various bacterial cultures found on food surfaces [126]. This necessitates the use of phage cocktails containing multiple phages targeting different bacterial strains. However, genetic engineering techniques can be used to modify phage host ranges and enhance their activity against specific target bacteria. Developing and optimizing phage cocktails can be complex and time-consuming. [114,122]. Thirdly, the potential for phage resistance development in target bacteria remains a critical issue. Continuous exposure to phages can drive the evolution of resistant bacterial strains, rendering the phage treatment ineffective [123]. Strategies to mitigate phage resistance, such as phage cocktails with diverse host ranges and the use of phage-resistant bacterial strains as indicators, are crucial for long-term success. Fourthly, ensuring consistent phage activity and efficacy in real-world packaging environments presents challenges [127]. Factors like the presence of organic matter, food matrix interactions, and the migration of phages from the packaging material to the food surface need to be carefully considered and addressed. Finally, regulatory hurdles related to the use of phages in food packaging exist [97]. Clear guidelines and regulations regarding phage production, characterization, application methods, and safety assessments are essential to facilitate the widespread adoption of phage-based packaging technologies.

## 5. Conclusions

Despite some ongoing challenges, bacteriophage application technology is increasingly recognized as a safe and effective method to completely eradicate or significantly reduce the presence of certain bacterial diseases in food. The diverse functional uses of bacteriophages in healthcare systems, the diverse prevention/control of food contamination, sustainable applications in food handling, storage, production, livestock and agriculture, disease management, and rapid detection of various methods have been used at various critical points. As mentioned previously, numerous countries have approved the commercialization of bacteriophage products, often leveraging established phage banks. These bacteriophages hold significant potential as a complementary tool within a multi-barrier approach to enhancing human well-being. The use of bacteriophages as antimicrobial tools in packaging applications, particularly in food, is a promising tool when food processors wish to eliminate only bacteria that can cause human disease while preserving the natural and often beneficial microbial community of food.

Antimicrobial active packaging technology has attracted much attention recently as it has the potential to stop/inhibit the growth of microorganisms without harming the food system and reducing the number of chemicals used. Antimicrobial food packaging films have emerged as a promising solution to extend the storage life of foods, as evidenced by numerous studies. These findings regarding bacteriophages as multifunctional antimicrobial agents have been satisfactory. While designing an active antimicrobial packaging film, it is advisable to address several important packaging parameters, such as antimicrobial efficacy, target performance, and the technical–mechanical and barrier abilities of the film. In addition, a series of validation tests should be performed as even small amounts of these antimicrobial agents can alter the packaging film properties in different ways. An information database should be created to understand the industrial feasibility of food packaging film design. Consumer awareness has increased, and their expectations for food quality have also increased. They expect appropriate food packaging and sustainability. Therefore, packaging development should consider raw material sources, biocompatibility, shelf life, food preservation mechanisms, disposal strategies, and environmental impact. Regulatory parameters, commercial viability, safety issues, economics, production capacity, sensory attributes, and, most importantly, customer/consumer acceptance are the key factors to consider when fortifying packaging with antimicrobial agents such as bacteriophages. Further research is needed on these factors. Foods that come into direct contact with active packaging must be safe. When developing safe and healthy active packaging materials, regulatory requirements must be met.

Harvesting full-scale biopolymers as antimicrobial carriers has also attracted significant attention due to safety and environmental concerns. However, unlike petroleum-based materials, biopolymers have limitations such as technical–mechanical stability and changes in mechanical properties. Therefore, it is necessary to establish a multidisciplinary approach based on research work conducted by various scientists around the world, bringing together experts in microbiology and biotechnology, especially in food packaging and technology, engineering, and materials science. A promising and sustainable future for the food packaging industry can be achieved by integrating bacteriophages as an adjunct to existing methods.

## Figures and Tables

**Figure 1 microorganisms-13-00401-f001:**
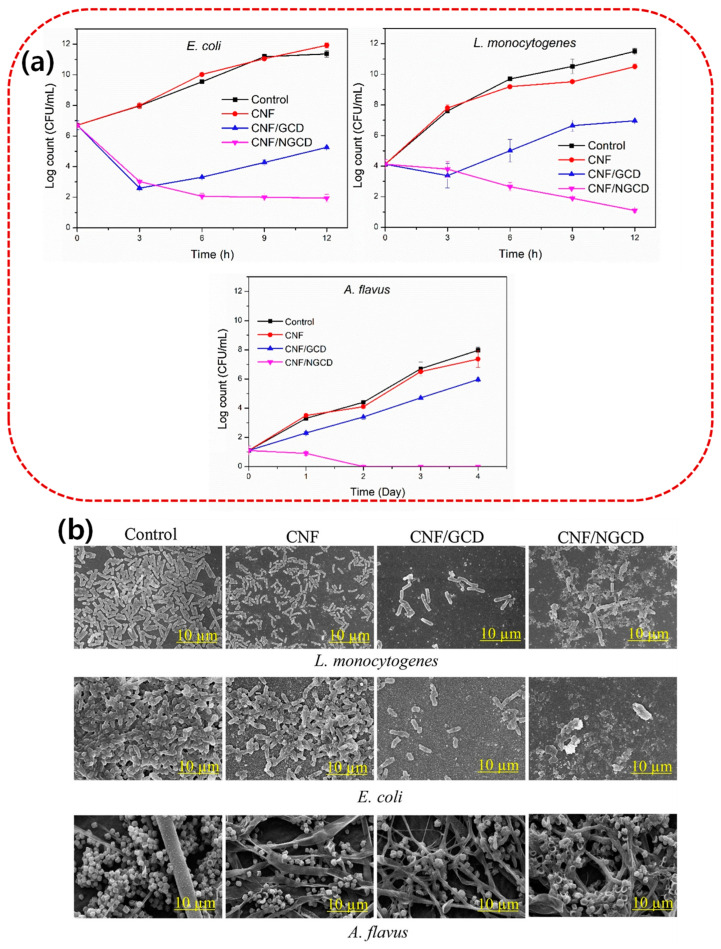
(**a**) Antimicrobial activity of the CNF-based films against *L. monocytogenes*, *E. coli*, and *Aspergillus flavus*. (**b**) SEM images of microorganisms treated with the CNF-based films. Adapted with permission from [39].

**Figure 2 microorganisms-13-00401-f002:**
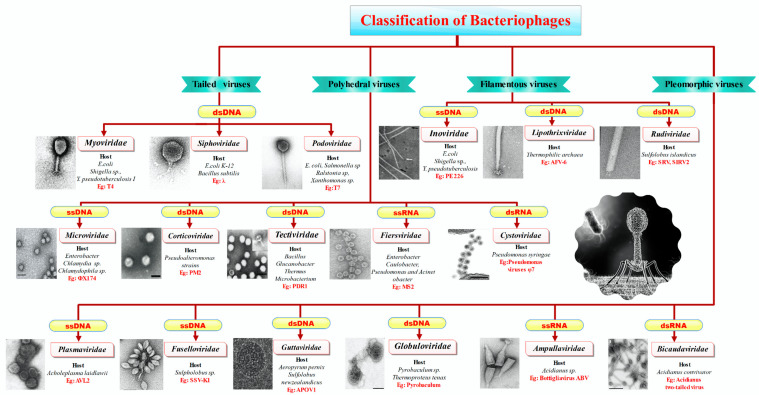
Classification, host, and examples of bacteriophages.

**Figure 3 microorganisms-13-00401-f003:**
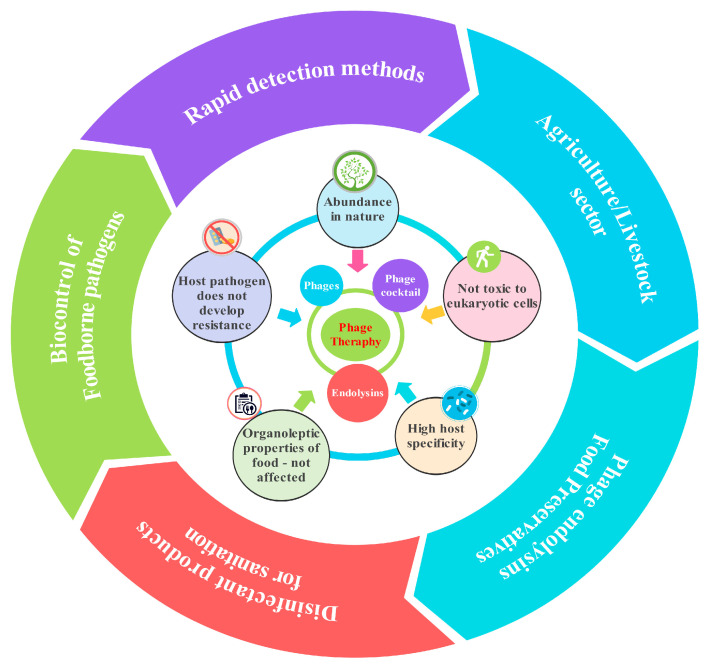
Schematic illustrations of the bacteriophage applications in various fields.

**Figure 4 microorganisms-13-00401-f004:**
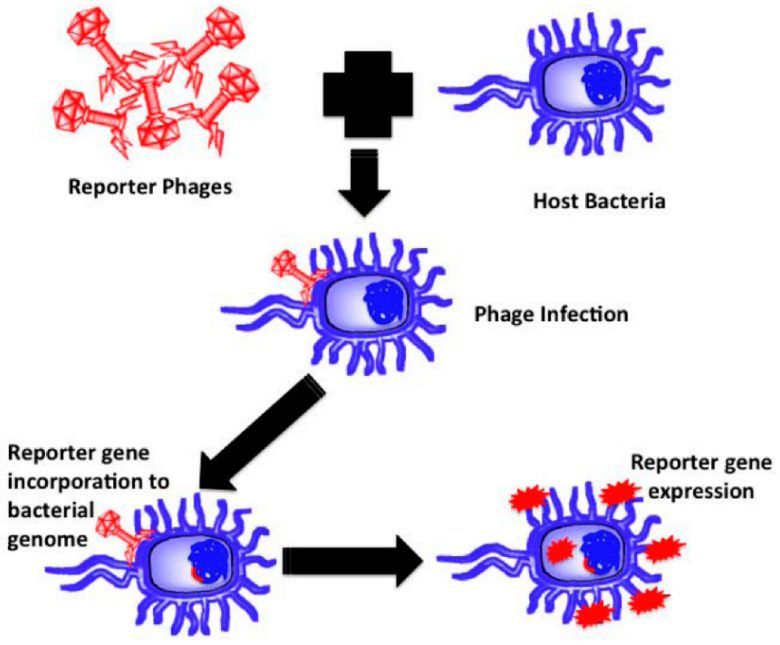
Diagram illustrating the basic idea of reporter phage-based detection of target pathogenic bacteria [105].

**Figure 5 microorganisms-13-00401-f005:**
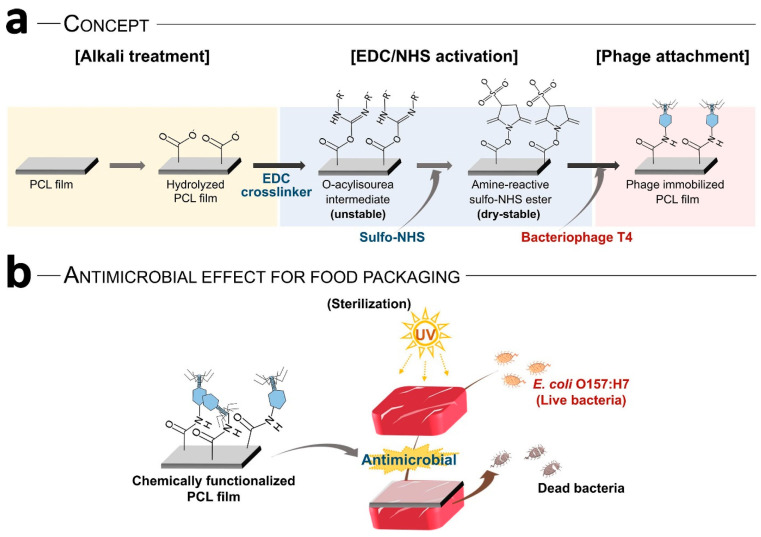
(**a**) The covalent bonding between the T4 phage and the functional group of the films and (**b**) phages added film for meat packaging applications [107].

**Figure 6 microorganisms-13-00401-f006:**
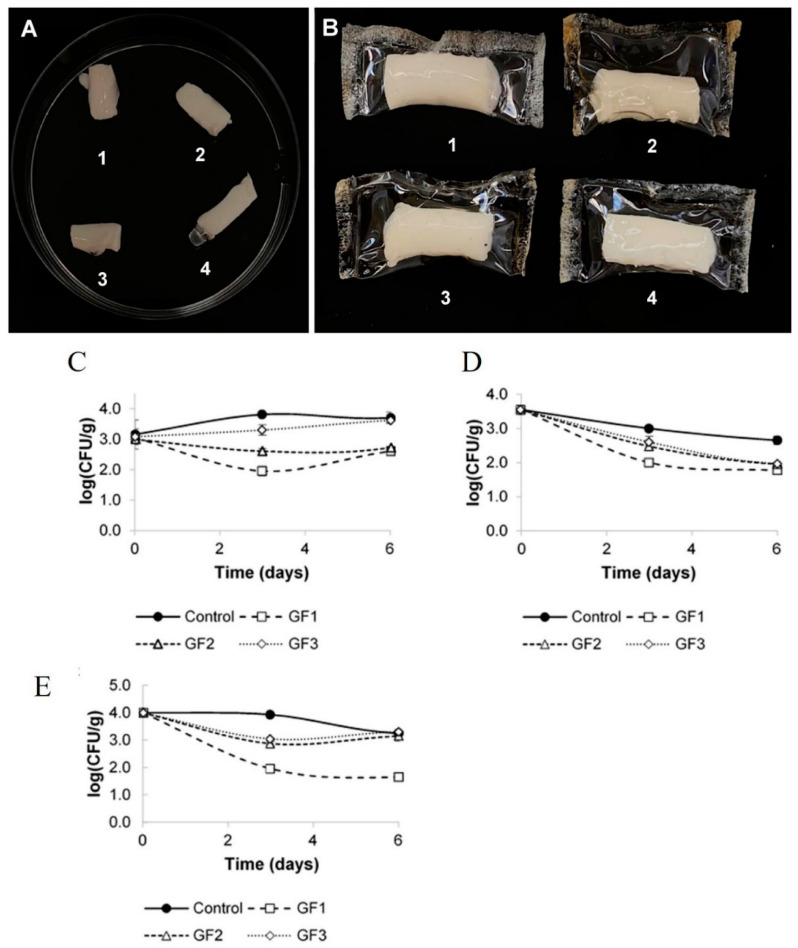
(**A**) Gelatine-coated cheeses and (**B**) film with a film-forming solution with different bacteriophage concentrations 1. Control (gelatine coating); 2. Gelatine coating or film prepared with a film-forming solution with a bacteriophage concentration of 1.75 × 10^8^ PFU/mL (GF1); 3. 1.16 × 10^8^ PFU/mL (GF2); 4. 6.35 × 10^7^ PFU/mL (GF3). Investigation of phage addition on *S. aureus* growth in cheese samples: (**C**) immersion of cheese pieces in the bacteriophage’s liquid medium at three different concentrations (GF1, GF2 and GF3), (**D**) coating the cheese samples with the film-forming solution at three different concentrations (GF1, GF2 and GF3), and (**E**) wrapping the cheese pieces with the dried films at three different concentrations (GF1, GF2 and GF3) [108].

**Table 1 microorganisms-13-00401-t001:** Summary of recently reported biopolymer-based nanocomposites for their antimicrobial properties and target applications.

Biopolymer Film	Functional Fillers	Target Microorganism	Application	Ref
PLA	Silver nanoparticles	*Bacillus subtilis*, *E. coli, Micrococcus luteus*, *Clostridium sporogenes*, and *Groenewaldozyma auringiensis*	Active packaging application	[42]
PEF	Ce–bioglass, ZnO, and ZrO_2_ nanoparticles	*E. coli* and *Staphylococcus aureus*	Active packaging application	[43]
GEL	*Vaccinium corymbosum* and derived carbon dots	*E. coli* and *L.monocytogenes*	Active and intelligent food packaging	[11]
CAR	Rose petals-derived carbon dots	*E. coli* and *L. monocytogenes*	Active and intelligent food packaging	[14]
CNF and WPI	Rosemary essential oil and titania nanoparticles	*Enterobacteriaceae*, *Pseudomonas* spp., *Lactobacillus*, *S. aureus*, *L. monocytogenes*, and *E. coli* O157: H7	Preserving lamb meat	[36]
Cellulose acetate	Montmorillonite-modified Cu^2+^	*E. coli*	Active food packaging	[37]
Cellulose nanofiber	*Brassica oleracea*-derived CDs	*E. coli* and *L. monocytogenes*	Active and intelligent food packaging	[38]
Alginate	Clay, essential oils (clove, cinnamon, and marjoram)	*E. coli*, *S. aureus*, and *L. monocytogenes*	Controlling pathogens in food packaging	[44]
CMC	Grape seed extract and ZnO	*E. coli* and *L. monocytogenes*	Storage life extension studies of meat (high-fat) products	[40]
Pectin	AgNPs	*E. coli* and *L. monocytogenes*	Active food packaging	[41]
Mineralized agar	Zn-minerals (Zn-phosphate, Zn-carbonate)	*E. coli*, *C. albicans*, and *S. aureus*	Food packaging	[45]
Alginate	Hydroxyapatite nanoparticles	*L. monocytogenes*	Fish and seafood packaging	[46]
Chitosan	Copper oxide	*E. coli*, *P. aeruginosa*, and *L. monocytogenes*	Water purification and food packaging	[47]
Chitosan	Titanium dioxide	*S. aureus*, *E. coli*, *Candida albicans*, and *Aspergillus niger*	Active food packaging	[48]
Cellulose	Quercetin	*E. coli* and *S. aureus*	Food packaging, environmental protection, and pharmaceutical industry application	[49]
Poly(lactide)/Poly(butylene adipate-co-terephthalate)	Clove and thyme essential oil	*S. aureus* and *E. coli*	Active food packaging	[50]
GEL/Agar	Sulfur quantum dots	*E. coli* and *L. monocytogenes*	Active food packaging	[51]
*Phyllanthus* *wightianus*	*P. wightianus* extract, flaxseed gel	*S. aureus* and *E. coli*	Meat product packaging (beef patties)	[52]
Chitosan/starch	Cellulose nanofibers and cinnamon essential oil	*S. aureus* and *E. coli*	Meat product packaging (raw beef meat)	[53]
Bacterial nanocellulose	Carbon dots	*E. coli* and *L. monocytogenes*	UV screening and forgery-proof packaging	[54]
Chitosan/GEL	Sulfur-functionalized chitin	*S. aureus*, *L. monocytogeneses*, *E. coli*, and *S. enterica*	Chicken meat preservation	[55]
GEL, cellulose	ZnO	*L. monocytogenes*, *E. coli*, and *S. aureus*	Active food packaging	[56]

CAR: carrageenan; CNF: cellulose nanofiber; WPI: whey protein isolate; PEF: poly(ethylene 2,5-furan dicarboxylate); PLA: poly(lactic acid); GEL: gelatin; and CMC: carboxymethyl cellulose.

**Table 2 microorganisms-13-00401-t002:** Summary of the bacteriophage application in the food packaging films/coatings.

Bacteriophage	Polymer Matrix	Packaging Type	Target Pathogen (s)	Food Application	References
T4	Polycaprolactone	Film	*E. coli* O157: H7	Raw beef	[107]
PhiIPLA-RODI	Gelatin	Film Coating	*S. aureus*	-	[108]
vB_EcoM34X vB_EcoSH2Q vB_EcoMH2W	Chitosan	Coating	*E. coli*	Tomatoes	[109]
ϕIBB-PF7A	Sodium alginate	Film	*P. fluorescens*	Chicken breast fillets	[110]
LISTEX™ P100	Cellulose membranes	Coating	*L. monocytogenes*	Ready-to-eat turkey breast	[111]
Felix O1	PHBV/PVOH PHBV/nanofiber	Coatings Film	*S. Enteritidis*	-	[112]
A511	WPC/pullulan	Film	*L. monocytogenes*	-	[113]
Cocktail (DT1 to DT6)	WPC	Film	*E. coli*	Meat	[114]
JN01	Gelatin	Film	*E. coli*	Beef	[115]
Felix O1 A511	Xanthan-coated polylactic acid	Film	*Salmonella* *L. monocytogenes*	Sliced turkey	[116]
*E. coli* O157: H7	Chitosan	Film	*E. coli*	Beef	[117]
CN8	Polyvinyl alcohol–whey protein isolate	Coating	*Clavibacter michiganensis*	Maize seeds	[118]
T7	Edible WPI	Coatings	*E. coli* *Vibrio* spp.	Fish feed pallets	[119]
*E. coli* O157: H7	Sodium alginate/polyethylene oxide nanofiber	Film	*E. coli*	Beef	[98]
Cocktail (*S. Enteritidis* F5-4, *S. Typhimurium* L2-1, and *S. Typhimurium* ICB1–1)	WPC, carboxymethyl cellulose, chitosan, sodium alginate	Coatings	*Salmonella*	Strawberries	[120]
A511	Poly (lactic acid) and whey protein/pullulan bilayer	Film	*L. monocytogenes*	Chicken breast	[121]
Cocktail (EC4 and φ135)	Sodium alginate	Film	*E. coli* and *Salmonella*	-	[122]
LISTEX™ P100	Sodium caseinate, sodium alginate mixed with gelatin, and polyvinyl alcohol	Film	*L. monocytogenes*	-	[111]

WPC: whey protein concentrate.

## Data Availability

Data are available in a publicly accessible repository.

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
