# Peer review of "The Role of Active Packaging in the Defense Against Foodborne Pathogens with Particular Attention to Bacteriophages"

_microorganisms, 2025, doi:10.3390/microorganisms13020401_

Round 1

Reviewer 1 Report

Comments and Suggestions for Authors

I highly appreciate the effort put into this review article, but there are substantial points in the review.

Keywords should be modified to suit the title of the review and the scope of the journal.

Section 2, "Antimicrobials in food packaging". This section does not fit entirely with the review topic and is best replaced by comparisons and clarification of the difference between the use of bacteriophages in packaging materials and conventional antimicrobials.

A lot of information in the review is restated in previous reviews and with the same references significantly, and it is better to reduce the re-information and focus on the new in the application of bacteriophage in the packaging materials.

Are there health or manufacturing challenges in the application of bacteriophage in packaging materials?

Defects, if any, from the application of bacteriophage in packaging materials should be clarified.

Author Response

Comments 1: Highly appreciate the effort put into this review article, but there are substantial points in the review.

Response 1: We are highly thankful to the reviewer for the positive comments. Suggested corrections have been incorporated and marked in the revised manuscript.

Comments 2. Keywords should be modified to suit the title of the review and the scope of the journal.

Response 2: Thank you for pointing this out. The keywords have been revised and modified to suit the manuscript title and the journal's scope (Line no 26)

Comments 3: Section 2, "Antimicrobials in food packaging". This section does not fit entirely with the review topic and is best replaced by comparisons and clarification of the difference between the use of bacteriophages in packaging materials and conventional antimicrobials.

Response 3: We are thankful for insightful comment.

            We would like to highlight that Section 2 has been introduced to understand active food packaging using antimicrobials. An attempt has been made to highlight the latest findings in antimicrobials used in food packaging (Line no 100)

            Section “4.3 Bacteriophages as antimicrobial agents in active food packaging” has been added to clarify the difference between the use of bacteriophages in packaging materials and conventional antimicrobials (Line no 414)

Comments 4: A lot of information in the review is restated in previous reviews and with the same references significantly, and it is better to reduce the re-information and focus on the new in the application of bacteriophage in the packaging materials.

Response 4: As suggested by the reviewer, the manuscript has been revised to reduce the re-information and attempt has been made to give more focus on the application of bacteriophage in the packaging materials (Section 4.3; Table 3). A new section, “Section 4.5 Challenges and defects in the application of bacteriophage in packaging materials,” has been added (Line no 563-586)

Comments 5: Are there health or manufacturing challenges in the application of bacteriophage in packaging materials?

Response 5: Yes, the application of bacteriophages in packaging materials poses health and manufacturing challenges.      

            Health Challenges: Despite their generally safe profile, rigorous quality control is necessary for phage preparations to ensure they are completely free from harmful bacteria and other contaminants. Rigorous quality control measures are necessary to avoid any potential health risks [1]. Clear and transparent communication about the safety and benefits of phage-based packaging is essential to address these concerns[2].

            Manufacturing Challenges: Maintaining phage viability and activity throughout the manufacturing process and during the shelf life of the packaged product can be challenging [3,4]. Factors like temperature, humidity, and the presence of other chemicals can affect phage stability [5,6]. Effectively incorporating phages into various packaging materials while maintaining their activity can be complex. Different packaging materials may require different approaches for phage delivery and immobilization [4]. Producing sufficient quantities of high-quality phage preparations for industrial-scale packaging applications can be costly and technically demanding.

            By addressing these challenges, phage-based packaging can offer a promising and sustainable alternative to traditional antimicrobial approaches in the food industry.

Comments 6: Defects, if any, from the application of bacteriophage in packaging materials should be clarified.

Response 6: As suggested, a new section, “Section 4.5 Challenges and defects in the application of bacteriophage in packaging materials,” has been added to discuss the challenges and defects of the application of bacteriophages in packaging materials. (Line No. 565-588).

Cited References:

  1. Jończyk-Matysiak, E.; Łodej, N.; Kula, D.; Owczarek, B.; Orwat, F.; Międzybrodzki, R.; Neuberg, J.; Bagińska, N.; Weber-Dąbrowska, B.; Górski, A. Factors Determining Phage Stability/Activity: Challenges in Practical Phage Application. Expert Rev Anti Infect Ther 2019, 17, 583–606, doi:10.1080/14787210.2019.1646126.
  2. Mohan, A.; Saluja, D.; Bajpai, U. Bacteriophages: The Natural Combatants to Fight AMR. In Emerging Paradigms for Antibiotic-Resistant Infections: Beyond the Pill; Springer Nature Singapore: Singapore, 2024; pp. 315–339.
  3. Narayanan, K.B.; Bhaskar, R.; Han, S.S. Bacteriophages: Natural Antimicrobial Bioadditives for Food Preservation in Active Packaging. Int J Biol Macromol 2024, 276, 133945, doi:10.1016/j.ijbiomac.2024.133945.
  4. Liu, D.; Van Belleghem, J.D.; de Vries, C.R.; Burgener, E.; Chen, Q.; Manasherob, R.; Aronson, J.R.; Amanatullah, D.F.; Tamma, P.D.; Suh, G.A. The Safety and Toxicity of Phage Therapy: A Review of Animal and Clinical Studies. Viruses 2021, 13, 1268, doi:10.3390/v13071268.
  5. López de Dicastillo, C.; Settier-Ramírez, L.; Gavara, R.; Hernández-Muñoz, P.; López Carballo, G. Development of Biodegradable Films Loaded with Phages with Antilisterial Properties. Polymers (Basel) 2021, 13, 327, doi:10.3390/polym13030327.
  6. Silva, Y.J.; Costa, L.; Pereira, C.; Mateus, C.; Cunha, Â.; Calado, R.; Gomes, N.C.M.; Pardo, M.A.; Hernandez, I.; Almeida, A. Phage Therapy as an Approach to Prevent Vibrio Anguillarum Infections in Fish Larvae Production. PLoS One 2014, 9, e114197, doi:10.1371/journal.pone.0114197.

Reviewer 2 Report

Comments and Suggestions for Authors

This review focuses on the research progress of bacteriophages and phagee-based materials and their potential applications, which has strong academic value. The research background and recent research progress of bacteriophage and phagee-based materials are introduced in detail, and it is recommended to receive after minor revision.

1.It is suggested to adjust Figure 4 and table 3 to appropriate positions and place them near the paragraphs describing their contents for readers' convenience.

2. Figure A, figure B and figure C, figure D and figure E in Figure 6 should be placed on one page.

3. The format of some references is not uniform. It is recommended to check carefully.

Author Response

Comments 1: It is suggested to adjust Figure 4 and table 3 to appropriate positions and place them near the paragraphs describing their contents for readers' convenience.

Response 1: As suggested by the reviewer, Figure 4 and Table 3 are placed in appropriate positions near the paragraphs describing their contents. (Line no 405 and 421).

Comments 2: Figure A, figure B and figure C, figure D and figure E in Figure 6 should be placed on one page.

Response 2: Figure 6 has been revised and placed on one page as recommended. (Line no 543).

Comments 3: The format of some references is not uniform. It is recommended to check carefully.

Response 3: All the references are revised and checked carefully. The errors have been rectified.

Reviewer 3 Report

Comments and Suggestions for Authors

Although the utilization of innovative bacteriophages as antimicrobial agents has gained substantial interest within various scientific community, there seems to be a lack of comprehensive literature review. In this respect, this review is timely and important for the scientific community. Because this review aims to provide detailed insights into 1) bacteriophages, including their classifications, mode of action, and multidisciplinary applications; and 2) developments of bacteriophage-based antimicrobial materials with their prospects in food packaging and food safety.

Overall, this review is well organized and well written. After publication, this review will serve as a resourceful guidelines for the further development of bacteriophage-based antimicrobial therapy.

Author Response

Comment 1: Overall, this review is well organized and well written. After publication, this review will serve as a resourceful guidelines for the further development of bacteriophage-based antimicrobial therapy.

Response 1: We are highly thankful to the reviewer for the positive comments and time.

Reviewer 4 Report

Comments and Suggestions for Authors

A very interesting manuscript on bacteriophages and their applications. The manuscript is well-written and is loaded with useful information. Attention is needed in the following:

1.       The manuscript needs extensive editing, especially regarding the use of scientific names. The first time a genus is mentioned, it should be written in full; then it can be abbreviated (e.g. full name in l. 156, abbreviation in l. 151). Scientific names should be written in italics (e.g. Table 1, Table 3, l. 912). In l. 314 it can be either pseudomonads, xanthomonads and vibrios (lowercase first letter, not in italics) or Pseudomonas, Xanthomonas and Vibrio (capitalized first letter and in italics).

2.       Table 2. Please use abbreviations in the table and explain them in a footnote.

3.       L. 204, It should read ‘Vibrio cholerae

4.       L. 365. It should read ‘lysogenic’

5.       L.412, ‘Typhimurium’ and ‘Enteritidis’ are serovars and should be written with capitalized the first letter and not in italics

Please apply the aforementioned changes to the whole text and not only to the few examples mentioned

Author Response

Comments 1: The manuscript needs extensive editing, especially regarding the use of scientific names. The first time a genus is mentioned, it should be written in full; then it can be abbreviated (e.g. full name in l. 156, abbreviation in l. 151). Scientific names should be written in italics (e.g. Table 1, Table 3, l. 912). In l. 314 it can be either pseudomonads, xanthomonads and vibrios (lowercase first letter, not in italics) or Pseudomonas, Xanthomonas and Vibrio (capitalized first letter and in italics).

Response 1: We are very thankful to the reviewer for their insightful comments. The scientific names have been revised as suggested. The changes made are highlighted in red text. (Line no 153, Table 2, Table 3, 204, 314)

Comments 2: Table 2. Please use abbreviations in the table and explain them in a footnote.

Response 2: As suggested, Table 2 has been revised (Line no 185 to 187).

Comments 3: L. 204, It should read ‘Vibrio cholerae

Response 3: Suggested changes have been made in the revised manuscript.

Comments 4: L. 365. It should read ‘lysogenic’

Response 4: We are thankful to reviewer for insightful comments. The suggested changes have been incorporated.

Comments 5: L.412, ‘Typhimurium’ and ‘Enteritidis’ are serovars and should be written with capitalized the first letter and not in italics

Response 5: As suggested, necessary changes have been made.

Round 2

Reviewer 1 Report

Comments and Suggestions for Authors

A commendable effort was made by the authors in making modifications and responding to inquiries.

Author Response

Comment 1: A commendable effort was made by the authors in making modifications and responding to inquiries.

Response 1: We are highly thankful to the reviewer for the positive comment.